# Preparation and Characterization of Guaiacol-Furfuramine Benzoxazine and Its Modification of Bisphenol A-Aniline Oxazine Resin

**DOI:** 10.3390/polym16060783

**Published:** 2024-03-12

**Authors:** Jing Wang, Riwei Xu

**Affiliations:** 1Basic Education School, Beijing Information Technology College, Beijing 100070, China; 2Beijing Key Laboratory of Electrochemical Process and Technology for Materials, Beijing University of Chemical Technology, Beijing 100029, China

**Keywords:** guaiacol, furfuramine, benzoxazine, bisphenol A-aniline oxazine

## Abstract

A new type of benzoxazine resin has been synthesized using a natural phenol source, guaiacol, and a biomass amines, furfuramine. The synthesis conditions were optimized; when the reaction molar ratio of guaiacol, furfuramine, and polyformaldehyde was 1:1:4, the highest synthetic yield was reached. The product was characterized via testing using transform infrared spectroscopy (FT-IR), gel permeation chromatography (GPC), mass spectrogram (MS), and nuclear magnetic resonance (^1^H-NMR) to confirm its molecular structure. A differential scanning calorimetry (DSC) test was conducted to analyze the thermodynamic properties of the product, and the results showed that the product decomposed and evaporated at around 180 °C, making it impossible to achieve self-curing. However, the prepared guaiacol-furfuramine benzoxazine resin (GFZ) can be blended and cured in certain proportions with bisphenol A-aniline oxazine resin (BAZ) as a GFZ/BAZ binary system (5:95, 10:90, 20:80, and 40:60). Dynamic mechanical analysis (DMA) test results showed that when the content of GFZ was 10%, the storage modulus of the copolymer resin was greatly improved. After conducting impact strength tests on the copolymer resin, it was found that the toughness of the copolymer resin had improved, and the maximum impact strength had increased by nearly three times. This indicates that the flexible long-chain structure in GFZ can effectively improve the toughness of the cured copolymer system. The reaction of active groups on benzoxazine molecules with other resins can not only improve the mechanical properties of their cured products, but also has important significance in the preparation of low-cost and environmentally friendly sustainable composite materials with excellent comprehensive performance.

## 1. Introduction

Benzoxazine resin is a new type of thermosetting resin developed on the basis of traditional phenolic resins. It is a six-membered heterocyclic system composed of oxygen and nitrogen atoms and is synthesized from phenolic compounds, aldehydes, and amine compounds. This heterocyclic intermediate can be ring-opening polymerized under heating or catalyst action to produce a network product-like phenolic resin structure which contains nitrogen. Benzoxazine compound (3,4-dihydro-3-substituted-2H-1,3-benzoxazine) is a kind of heterocyclic compound dehydrated-synthesized from phenols, amine compounds, and formaldehyde under certain conditions [1,2]. Compared with traditional phenolic resins, benzoxazine resin has significant advantages. Due to its three-dimensional network structure formed through self-ring-opening polymerization, benzoxazine does not release small molecules during the curing process, resulting in low porosity, near “zero” volume shrinkage, low stress, and few micro-cracks. In addition, benzoxazine resin is a low viscosity cyclic monomer before ring-opening polymerization, with good solubility and excellent process performance. Therefore, it can be used as a matrix material for preparing composite materials. According to its structural formula, benzoxazine resin has a flexible molecular design, and different phenolic or amine sources can be selected according to application needs in order to synthesize different structures of benzoxazine [3,4,5].

Burke and Bishop [6] reported that benzoxazine monomers can undergo carbon-nitrogen nucleophilic substitution reactions with different types of compounds with active hydrogen (HY), proposing the ring-opening mechanism for the first time and pointing out that this amino alkylation ring-opening reaction does not release volatile substances.

Tom Higginbot [7] reported that active amides containing primary and secondary amine groups can also initiate the ring-opening polymerization of benzoxazine monomers. Dunkers [8] pointed out that the ring-opening polymerization reaction of 3,4-dihydro-3,6-didisubstituted-2H-1,3-benzoxazines monomer can be successfully achieved by using trifluoroacetic acid, sebacic acid, and p-cresol as catalysts.

Hyun et al. [9] introduced acetylene into benzoxazine monomer, opening up a new pathway for ring-opening polymerization. Under heating conditions, both an acetylene-based polyaddition reaction and benzoxazine ring-opening polymerization can occur, reducing the curing temperature. At present, experts are still studying the ring-opening mechanism in the hope of gaining a deeper theoretical understanding of the curing reaction of benzoxazine monomers [10].

In the study of new thermosetting resins, it was found that monofunctional benzoxazine undergoes a chain-transfer reaction during ring-opening polymerization, resulting in a low molecular weight product, which limits its application range. However, bifunctional benzoxazines (such as bisphenol A type) can be used as high-performance materials. Due to their own molecular structures, this type of benzoxazine polymer has disadvantages, including low crosslinking density, brittleness, and poor toughness. Therefore, in order to meet the requirements for special use, it is necessary to modify benzoxazine appropriately [11,12,13].

Grabarnic [14,15] used benzoxazine as a curing agent for phenolic resins and studied the curing reaction of monofunctional benzoxazine. It was found that the average molecular weight of the heat-cured product was only about 1000. At the same time as the chain grew, there was a thermal decomposition reaction of monomers, which made it impossible to obtain high molecular weight linear polymers.

Ishida et al. [16] used poly ε-caprolactone (PCL) to blend with benzoxazine, and it was found that the two could be mixed at lower temperatures. Due to the inter-molecular hydrogen bonding between the phenolic hydroxyl group of polybenzoxazine and the carbonyl group of PCL, PCL can be mixed with benzoxazine to a certain extent; it can accelerate the ring-opening of benzoxazine, but delay its polymerization, ultimately leading to more complete polymerization of benzoxazine. Adopting PCL to modify benzoxazine can improve its toughness, with minimal impact on thermal stability and mechanical properties [17].

Guaiacol is a plant chemical substance found in guaiacol trees. It is mainly present in guaiacol resin or pine oil, as well as in creosote obtained from wood distillation. Its structure comes from benzyl ether and phenol [18]. Furfuramine, as a biomass amine, can replace petroleum based raw materials [19]. This paper used these two natural materials to synthesize a new type of benzoxazine: guaiacol-furfuramine benzoxazine resin (GFZ). We also studied the modification of this product to bisphenol A-aniline oxazine resin (BAZ).

Based on the excellent molecular design of benzoxazine, other reactive groups can be introduced to the structure, which can not only improve the mechanical properties of its cured copolymer, but also have important significance when preparing low-cost and environmentally friendly sustainable composite materials that exhibit excellent comprehensive performance [20]. In this experiment, the biomass materials guaiacol and amine were used to instead of bisphenol A and aniline to produce a high-performance bio-based benzoxazine composite material, which could provide guidance for researchers in replacing petroleum-based raw materials with low-cost and pollution-free biomass materials directly obtained from nature.

## 2. Materials and Methods

### 2.1. Material Preparation

#### 2.1.1. Preparation of GFZ

200 mL of toluene solution (AR, Beihua Fine Chemicals Co., Ltd., Beijing, China) was added into a 500 mL three-necked flask equipped with a condenser and stirrer. Then, 31 g (0.25 mol) of guaiacol (Beijing Chemical Reagent Co., Ltd., Beijing, China) and 24.25 g (0.25 mol) of furfuramine (Beijing Chemical Reagent Co., Ltd., Beijing, China) were added to the flask sequentially. Following this, 30 g (0.5 mol) of paraformaldehyde (AR, Tianjin Chemical Reagent Research Institute, Tianjin, China) was slowly added under strong stirring in an ice-water bath. After being stirred vigorously at room temperature to mix the reactants evenly, the mixture reacted for 2 h in a 95 °C oil bath. Upon completion of the reaction, 1 mol/L sodium hydroxide solution (AR, Beijing Century Red Star Chemical Co., Ltd., Beijing, China) was added and the mixture was stirred. Then, the mixture was poured into a separating funnel to separate the product, and sodium hydroxide solution was used to wash it continuously. Deionized water was used next to wash the product until neutrality, and the product was finally poured onto a porcelain plate and vacuum-dried at 60 °C for 24 h to obtain a white solid (Figure 1).

#### 2.1.2. Preparation of BAZ

500 mL of toluene solution was first added into a 1000 mL three-necked flask equipped with a condenser and stirrer, and then 14 g (0.5 mol) of bisphenol A and 120 mL (1 mol) of aniline were added sequentially. Following this, 60 g (2 mol) of paraformaldehyde was slowly added under strong stirring in an ice-water bath. After being stirred vigorously at room temperature to mix the reactants evenly, the mixture reacted in an oil bath at 105 °C for 2 h. After the reaction was complete, 1 mol/L alkaline solution was added and stirred. Then, the mixture was poured into a separating funnel to separate the product. The alkaline solution and deionized water were subsequently used to wash the product until neutrality was reached. Then, the product was poured onto a porcelain plate and vacuum-dried at 60 °C for 24 h to obtain a light yellow solid (Figure 2).

#### 2.1.3. The Co-Curing of GFZ/BAZ Copolymer

GFZ was mixed and cured with bisphenol A (AR, Beijing Haidian Xinxing Reagent Factory, Beijing, China) at mass ratios of 5:95, 10:90, 20:80, and 40:60, respectively. A solution blending method was adopted in this process, and different ratios of bisphenol A and GFZ were separately dissolved in a certain amount of tetrahydrofuran solution (AR, Beijing Beihua Fine Chemicals Co., Ltd., Beijing, China) and stirred for 5 h. The solvent in the system was vacuum-extracted to obtain the target system. Certain amounts of the above blended systems were separately placed into a mold and cured in a vacuum oven with the curing process at 120 °C for 2 h, 140 °C for 2 h, 160 °C for 2 h, 180 °C for 3 h, and 200 °C for 2 h.

### 2.2. Characterization

The structure of GFZ was analyzed via testing instruments as follows:

Transform infrared spectroscopy (FT-IR) (Nicolet 60 SXB Fourier; Thermo-Nicolet, Inc., Waltham, MA, USA). Scanning range: 400–4000 cm^−1^; KBr compression.

Gel permeation chromatography (GPC) (GPC 515-2410 system; Waters, Milford, MA, USA). Column temperature: 30 °C; mobile phase: tetrahydrofuran (THF); concentration: 2 mg/mL; flow rate: 1 mL/min; chromatographic column: Styragel HT3-HT5-6E (in series).

Mass spectrometry (MS) (Waters Quattro Premier XE, Waters, USA). Cone hole voltage: 30 V.

^1^H-NMR (Bruker Avance 600 MHZ NMR instrument; Bruker Switzerland AG, Fallanden, Switzerland). Solvent: CDCl_3_; internal standard: Si(CH_3_)_4_; tested at room temperature.

The thermal performance of GFZ was tested using instruments as follows:

Differential scanning calorimetry (DSC) (Perkin Elmer Pyris 1 thermal analyzer; Perkin Elmer Enterprise Management (Shanghai) Co., Ltd., Shanghai, China). Environment: N_2_; heating rate: 10 °C/min; reference material: Al_2_O_3_; testing temperature range: room temperature-350 °C.

The dynamic thermo-mechanical properties of the GFZ/BAZ copolymer were characterized via use of instruments as follow:

Dynamic mechanical analysis (DMA) (DMA 242 E; NETZSCH Scientific Instrument Trading (Shanghai) Co., Ltd., Shanghai, China). Testing temperature range: room temperature-250 °C; heating rate: 5 °C/min; sample size: 25 × 6.5 × 1.5 mm^3^; testing frequency: 1 Hz.

The impact strength of GFZ/BAZ copolymer was tested using instruments as follows:

A simple supported beam pendulum impact testing machine (RESIN IMPACTOR (P/N 6957.000; Dongguan Municipal People’s Instrument and Equipment Co., Ltd., Guangzhou, China). Span: 70 mm; spline size: 80 × 10 × 4 mm^3^; ten splines at each mass ratio [21].

## 3. Results and Discussion

### 3.1. Determination of Synthesis Conditions

#### 3.1.1. The Effect of Solution on Yield

According to the process in Section 2.1.1, different solvents (toluene, chloroform, n-butanol, and dimethylformamide (DMF)) were separately used in the reaction, and the yield of GFZ for each solvent is shown in Table 1. It can be seen that when the dielectric constant of the solvent was higher than four, there was no oxazine ring generated in the product. It was observed that the polarity had no direct effect on the yield, so toluene was chosen as the most suitable solvent for this experiment, also considering cost and post-treatment issues. However, considering environmentally friendly materials, solvent-free methods, or low toxicity solvents as alternatives should be further studied.

#### 3.1.2. The Effect of Aldehyde Types on Yield

Under the same reaction conditions, polyformaldehyde, trimeric formaldehyde, and formaldehyde aqueous solution (0.2 mol each) were separately used as reactants. The yield of GFZ for each aldehyde type is shown in Table 2. It can be seen that when polyformaldehyde was used, the product yield was the highest, so polyformaldehyde was selected as the experimental reactant.

#### 3.1.3. The Effect of Aldehyde Amount on Yield

Under the same conditions, the amount of formaldehyde (paraformaldehyde) used was tested at 0.2 mol, 0.3 mol, 0.4 mol 0.5 mol and 0.6 mol. The yield of GFZ under each condition is shown in Table 3. The higher the amount of paraformaldehyde was, the higher the yield became. However, when the amount of paraformaldehyde reached a certain level, the yield increase became insignificant. Therefore, in order to save raw materials, the amount of paraformaldehyde for use was selected as 0.4 mol. The best ratio of paraformaldehyde to guaiacol is 4:1, as shown in Figure 1.

In general, the optimal synthesis conditions are as follows: under reflux (about 90–100 °C), reaction time of 2 h, toluene is used as the solvent, and a reaction molar ratio of raw materials, i.e., guaiacol, furfuramine, and polyformaldehyde of 1:1:4, taking into account the optimal yield and cost.

### 3.2. Testing and Characterization of GFZ

The designed structural formula of GFZ can be seen below (Figure 3). To further demonstrate the successful synthesis of this benzoxazine in the experiment, a series of tests and characterizations were conducted on the product.

As shown in Figure 2, which represents the FT-IR of GFZ, a characteristic peak in the oxazine ring appears at 917.62 cm^−1^ (the similar structure of oxazine composite can be seen in Figure A1, Appendix A), and there is no hydroxyl stretching vibration absorption peak near 3427 cm^−1^, indicating that the phenolic hydroxyl group has been fully reacted.

The other infrared peaks appearing in the figure are described as follows: the peak at 3089 cm^−1^ is attributed to the antisymmetric stretching vibration peak of the C=C bond on the allyl group; the peak at 2921 cm^−1^ is attributed to the CH_3_ anti symmetric stretching vibration peak; the peak at 1614 cm^−1^ is attributed to the C=C bond stretching vibration peak on the allyl group; the skeleton vibration peaks of the benzene ring are located near 1583 and 1501 cm^−1^; the peak at 1353 cm^−1^ is attributed to the symmetric deformation peak of the methyl group; and the peak 1264 cm^−1^ is attributed to the rocking peak of methylene in the oxazine ring. There is almost no absorption peak around 3400 cm^−1^, indicating a very low content of oligomers.

In order to analyze the molecular weight of the product, GPC and MS tests were conducted. Figure 3 shows the GPC spectrum of GFZ. It can be seen that the molecular weight distribution of the product is narrow, concentrated around 100–300.

Figure 4 shows the mass spectrum of GFZ. The molecular weight of GFZ obtained is 245.9, while its theoretical molecular weight is 245.1, which is basically consistent (Figure 4). The product that appears at 234 is analyzed as:

Figure 5 shows the ^1^H-NMR spectrum of GFZ, and each proton peak in the figure can be identified via the chemical shift (the original spectrum with detailed data can be found in Figure A2, Appendix A). The peak at 6.756–6.853 ppm (a,b,c) can be attributed to the proton peak on the benzene ring; the peak at 6.245–6.325 ppm (d,e) can be attributed to the proton peak of -CH on the furan ring; the peak at 7.408 ppm (f) can be attributed to the proton peak of -O-CH= on the furan ring; the peak at 4.987 ppm (g) can be attributed to the proton peak of -O-CH_2_-N-; the peak at 3.948–4.016 ppm (h,i) can be attributed to the proton peak of -Ar-CH_2_-N-; and the peak at 3.885 ppm (j) can be attributed to the proton peak of -O-CH_3_. The area ratio of each peak, a:b:c:d:e:f:g:h:i:j, is equal to 1.099:1.023:1.000:1.001:1.024:1.003:2.103:2.189:2.155:3.113, and the theoretical ratio of proton numbers, a:b:c:d:e:f:g:h:i:j, is equal to 1:1:1:1:1:1:2:2:2:3.

From the data above, it can be seen that the chemical shifts of hydrogen atoms at various positions of the product can infer the positions of each hydrogen in the product, and its peak area ratio is consistent with the theoretical value ratio. It can be inferred that the final product of the reaction is the designed target product (the similar structure of oxazine composite can be seen in Figure A3, Appendix A).

Figure 6 shows the DSC curing curve of GFZ. It can be seen that the melting point of the monomer is 81.4 °C, and a broad endothermic peak appears around 218 °C. No curing exothermic peak appears in the entire DSC spectrum, and the monomer decomposes at 250 °C, indicating that the monomer of this pure oxazine cannot be cured. The experimental results show that pure oxazine monomers cannot be self-cured. Because hydrogens in the ortho and para positions have been replaced, the solidification points have decreased, and the hydrogen at the β position lowers the ring-opening temperature, promoting ring opening. When the hydrogen at the β position is replaced, the product reaches its boiling point and evaporates before solidification [22]. Because of this drawback in the structure, in order to solve the problem of non-self-curing, furfuramine may be alternated with a kind of long chain amine without a furan ring. The GFZ resin prepared in this paper can be blended with other curable resins or catalysts to reduce the curing temperature for applications.

BAZ’s DSC curing curve is shown in Figure 7. It is obvious that there are two peaks in the entire curve. The first peak is the melting endothermic peak of bisphenol A, and the second peak is the curing exothermic peak of BAZ. The initial curing temperature is around 243 °C, and the peak curing temperature is around 252 °C. The shape is sharp, indicating a fast curing reaction rate.

### 3.3. Testing and Characterization of GFZ/BAZ Copolymer

The flexible long carbon chain structure in GFZ enables it to be blended and cured with BAZ, thus improving the brittleness of BAZ. In order to study the mechanical properties of the GFZ/BAZ copolymer system, DMA tests were conducted using cured copolymer samples of different proportions, and the results are shown in Figure 8, which studies the relationship between the loss factor of GFZ/BAZ copolymer and temperature. It can be seen that after adding GFZ to BAZ, the peak of the loss factor increases, while the temperature range narrows, and the peak shifts towards a higher temperature. There is only one peak on the curve, indicating that the GFZ/BAZ copolymer has a good copolymerization effect, and no phase separation phenomenon occurred.

Figure 9 shows the relationship between temperature and the storage modulus of GFZ/BAZ copolymer spline samples with different ratios. When GFZ content reaches 10%, the storage modulus of the copolymer greatly improves, which means that the strength of the material is greatly enhanced. When the content of GFZ continuously increases, the storage modulus decreases again. The GFZ and BAZ prepared in this paper are both small molecule complexes with oxazine rings and π-π conjugated structures. Due to the similarities in the molecular structures of these two complexes, the two resins also have good compatibility. During the copolymerization process of GFZ and BAZ, the oxazine ring in GFZ opens and enters the cross-linking network, while the oxazine ring and furan ring in BAZ also participate in the curing process, then enter the cross-linking network, making it easy for the two to blend and solidify. Meanwhile, the long carbon chains contained in GFZ molecules increase the flexibility of the molecular chains while also reducing the crosslinking density, which results in a decrease in the rigidity of the crosslinking network and a decrease in the storage modulus.

The impact strength test and replicated test were conducted on the GFZ/BAZ copolymers, and the results are shown in Table 4 and Figure 10. The addition of GFZ to BAZ for blending and curing effectively improved the impact strength compared with the pure BAZ. When the content of GFZ in the copolymer resin reached 10%, the impact strength of the copolymer resin increased nearly three times compared to the original, indicating that the addition of GFZ can improve the toughness of the copolymer resin. This is because there are many long straight carbon chains among the molecules, and the longer the carbon chains are, the softer the molecular chain will become, and the better the toughness of the cured material [23]. The addition of long aliphatic chain segments in GFZ has a good toughening effect on the copolymerization system.

## 4. Conclusions

This paper synthesized a novel benzoxazine from natural materials: guaiacol and furfuramine. The synthesis process was optimized and the target product was tested and characterized using FT-IR, GPC, MS, and ^1^H-NMR. The results showed that it was the desired benzoxazine, and the structure was confirmed. At the same time, DSC testing was conducted on the product, and the results showed that the melting point of the product was around 81 °C, while at around 180 °C, the product decomposed and evaporated, making it impossible to achieve self-curing. The product was also co-mixed and cured with bisphenol A-aniline oxazine resin at mass ratios of 5:95, 10:90, 20:80, and 40:60. The copolymer was subjected to DMA testing, and it was concluded that when the content of GFZ reached 10%, the energy storage modulus of the copolymer was greatly improved, indicating that this new type of benzoxazine can effectively enhance the strength of the copolymer. After conducting impact strength tests on the GFZ/BAZ copolymer, it was found that the toughness of the cured composite material had been improved, and the maximum impact strength could be increased to three times that of the original value. This indicated that the new benzoxazine can effectively enhance the toughness of GFZ/BAZ copolymer. The new benzoxazine synthesized in this experiment not only effectively improved the mechanical properties of the material, but also has the potential to improve the impact toughness of benzoxazine resin in composite materials. However, due to its inability to self-cure, it is recommended to copolymerize with other types of benzoxazine for modification during application. In addition, the biomass phenol source guaiacol was used instead of bisphenol A, along with a biomass amine instead of aniline. This approach can effectively solve the problems of shortage of petroleum resources and increasingly serious pollution, and encourage people to seek low-cost, green, and recyclable biomass materials that can be directly obtained from nature to replace petroleum-based raw materials. However, compared to traditional benzoxazine, bio-based benzoxazine still needs to be improved in terms of performance, including optimization of the production process and environmental protection, which require further research. Therefore, the industrialization of bio-based benzoxazine still has a long way to go.

## Data Availability

Data available on request due to privacy.

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
