# Peer review of "Preparation and Characterization of Guaiacol-Furfuramine Benzoxazine and Its Modification of Bisphenol A-Aniline Oxazine Resin"

_polymers, 2024, doi:10.3390/polym16060783_

Round 1

Reviewer 1 Report

Comments and Suggestions for Authors

1- It is suggested to add quantitative results in one or two lines to the abstract.

2- It is suggested that figures 1 and 2 be removed from the introduction section. In addition, the innovation of this research should be clearly stated at the end of the introduction.

3- The solvents used in this research are toxic and dangerous for humans. What suggestions do the authors have for changing the solvents? Does the polarity of the solvent have an effect on the final yield?

4- It seems that according to the structure, using the Raman method is better. If the authors can perform this test on the prepared samples.

5- Are the prepared resins suitable for film formation? What conditions do you suggest for this purpose?

Author Response

Dear reviewer,

Reviewer 2 Report

Comments and Suggestions for Authors

1. The paper lacks clarity in detailing the optimized synthesis conditions of the benzoxazine resin from Guaiacol and Furfuramine. Without a clear explanation, it's difficult to assess the reproducibility and reliability of the synthesis process.

2. The characterization techniques employed, such as FT-IR, GPC, MS, and 1H-NMR, are standard, but the paper fails to provide sufficient detail on the specific parameters used in each analysis. This omission raises questions about the rigor of the characterization process.

3. The assertion that the product is the desired benzoxazine and the structure was confirmed lacks adequate support without detailed spectral data and analysis provided in the paper.

4. The claim of improved mechanical properties lacks comprehensive evidence. While DMA and impact strength tests were conducted, the paper fails to provide statistical analysis or replicate the experiments to establish the reliability of the results.

5. The discussion on the copolymerization with bisphenol A-aniline oxazine resin lacks depth. The paper does not explore the mechanism of copolymerization or discuss the compatibility of the two resins, which is crucial for understanding the performance enhancement observed.

6. The statement regarding the potential aerospace applications of the synthesized benzoxazine resin lacks substantiation. Without experimental data or comparative analysis with existing aerospace materials, this claim appears speculative.

7. The decomposition and evaporation of the product at around 180 ℃, hindering self-curing, is a significant drawback that undermines the practical utility of the resin. The paper should explore potential strategies to mitigate this issue or discuss alternative curing methods.

8. The paper lacks discussion on the environmental impact of the synthesis process. Given the emphasis on using natural materials and environmentally-friendly materials, it's essential to address any potential environmental concerns associated with the synthesis process.

9. The conclusion regarding the effective utilization of energy and development of environmentally-friendly materials lacks depth. The paper should provide a more thorough analysis of the environmental benefits of using natural products and discuss potential challenges or limitations in scaling up production.

10. Overall, the paper lacks a critical evaluation of the limitations and challenges associated with the synthesized benzoxazine resin. A more balanced discussion of both the strengths and weaknesses of the material would provide a more comprehensive understanding of its potential applications.

Author Response

Dear reviewer,

Round 2

Reviewer 2 Report

Comments and Suggestions for Authors

The article can be accepted